# Attenuation of *Aeromonas hydrophila* Infection in *Carassius auratus* by YtnP, a *N*-acyl Homoserine Lactonase from *Bacillus licheniformis* T-1

**DOI:** 10.3390/antibiotics10060631

**Published:** 2021-05-26

**Authors:** Mengfan Peng, Wentao Tong, Zhen Zhao, Ling Xiao, Zhaoyue Wang, Xuanming Liu, Xuanhao He, Zengfu Song

**Affiliations:** 1National Demonstration Center for Experimental Fisheries Science Education, Shanghai Ocean University, Shanghai 201306, China; pengmengfan@sina.cn (M.P.); a18621313842@163.com (W.T.); caomuzhen@163.com (Z.Z.); enid_01@126.com (L.X.); wangzzzzzyue@163.com (Z.W.); xml_dhou@163.com (X.L.); hxh1979199774@163.com (X.H.); 2National Pathogen Collection for Aquatic Animals, Shanghai Ocean University, Shanghai 201306, China

**Keywords:** YtnP, *Bacillus licheniformis*, *Aeromonas hydrophila*, anti-virulence

## Abstract

In this experiment, the quorum quenching gene *ytnP* of *Bacillus licheniformis* T-1 was cloned and expressed, and the effect against infection of *Aeromonas hydrophila* ATCC 7966 was evaluated in vitro and vivo. The BLAST results revealed a 99% sequence identity between the *ytnP* gene of T-1 and its homolog in *B.*
*subtilis* sub sp. BSP1, and the dendroGram showed that the similarity in the YtnP protein in T-1 was 100% in comparison with *B.*
*subtilis* 3610, which was categorized as the Aidc cluster of the MBL family. The AHL lactonase activity of the purified YtnP was detected as 1.097 ± 0.7 U/mL with C6-HSL as the substrate. Otherwise, purified YtnP protein could significantly inhibit the biofilm formation of *A.*
*hydrophila* ATCC 7966 with an inhibition rate of 68%. The MIC of thiamphenicol and doxycycline hydrochloride against *A. hydrophila* reduced from 4 μg/mL and 0.5 μg/mL to 1 μg/mL and 0.125 μg/mL, respectively, in the presence of YtnP. In addition, YtnP significantly inhibited the expression of five virulence factors *hem, ahyB, ast, ep, aerA* of *A. hydrophila* ATCC 7966 as well (*p* < 0.05). The results of inhibition on virulence showed a time-dependence tendency, while the strongest anti-virulence effects were within 4–24 h. In vivo, when the YtnP protein was co-injected intraperitoneally with *A. hydrophila* ATCC 7966, it attenuated the pathogenicity of *A. hydrophila* and the accumulated mortality was 27 ± 4.14% at 96 h, which was significantly lower than the average mortality of 78 ± 2.57% of the *Carassius auratus* injected with 10^8^ CFU/mL of *A. hydrophila* ATCC 7966 only (*p* < 0.001). In conclusion, the AHL lactonase in *B. licheniformis* T-1 was proven to be YtnP protein and could be developed into an agent against infection of *A. hydrophila* in aquaculture.

## 1. Introduction

The occurrence of bacterial diseases in the aquaculture industry has led to rising demand for biocontrol agents as an alternative to antibiotics. Antibiotics act by disrupting processes such as bacterial cell wall synthesis, DNA proliferation, and protein synthesis, but the abuse of antibiotics results in resistance of bacteria [1]. Therefore, discovering a new type and effective antibiotic alternatives has become an emergent task for researchers. As we know, with the assistance of the virulence factor, pathogens can enter a host and break its immune defense mechanism. Previous studies have shown that pathogenic virulence factors are regulated by the quorum sensing (QS) system, through which information communication between them could be blocked to reduce the pathogenicity of the bacteria [2,3]. Quorum quenching (QQ) interferes with the quorum sensing system to stop the infection by pathogens through inhibiting the synthesis and accumulation of signal molecules or by enzymatic degradation of signal molecules. Therefore, an anti-virulence strategy will become a potential approach against infections in the future. 

*N*-acyl-homoserine lactone (AHL), termed autoinducers, is one of the Quorum Sensing signaling molecules, which is secreted by the cell itself to manipulate the expression of the virulence gene. As a result, the AHL-degrading enzymes against microbial infections have attracted more intention in recent years [4,5]. AHL-mediated quorum sensing (QS) is widely present in Gram-negative bacteria and plays a key role in the physiological state of bacteria. Currently, the AHL mediated QS system has been studied, which is considered as a promising method to regulate the nitrogen metabolism process [6]. To date, approximately 30 types of AHL lactonases have been determined [7], but there have been few reports about YtnP as a potential QQ gene and YtnP as an *N*-acyl-homoserine lactonase. 

The earliest report on the *ytnP* gene originated from the whole genome sequence of *B. subtilis*, in which it was annotated as an “unknown gene” among the 4100 genes identified [8]. Studies on the properties and functions of YtnP were only conducted by Schneider et al. [9], which indicated that YtnP could block the signaling pathways to interfere with the biofilm formation in *P. aeruginosa* [10]. In our previous study, a *Bacillus* sp. strain isolated from freshwater showed the quorum quenching activities [11]. To analyze the T-1 draft genome by the Illumina Hiseq 2500 high-throughput sequencing platform, the quorum quenching gene *ytnP* was discovered and amplified. Until now, that was the first report published on the QQ enzymes of YtnP in *B. licheniformis*. 

*Aeromonas hydrophila* is found in freshwater and seawater, mainly in freshwater culture environments, and causes a wide variety of symptoms in fish, including tissue swelling, necrosis, ulceration, and hemorrhagic septicemia [12,13,14]. The previous studies indicated there were two QS systems in *A. hydrophila*; the first signaling molecule is AHL, and the second signal molecule is AI-2. AHL was identified as the signal molecule to regulate the expression of virulence factors, biofilm maturation, and the type II, III, and VI secretion systems [15,16,17,18,19]. Hence, in this present study, we set the *A. hydrophila* ATCC 7966 as the research target strain and try to explore the anti-virulence effect of YtnP protein from *B. licheniformis* T-1 against infection of *A. hydrophila* in *Carassius auratus*, which will help us to understand the function of QQ enzyme and the relationship among the microorganism ecology system, and develop the new biological agents to dissolve the drug-resistant problem in the way of QQ. 

## 2. Materials and Methods

### 2.1. Strains and Culture Conditions

*B. licheniformis* T-1 was isolated from the freshwater aquaculture pond at Binhai of Shanghai, China. Strain T-1 was preserved in the National Pathogen Collection Center for Aquatic Animals (Shanghai, China) under the registered number of KP117098. The strain was activated at 30 °C for 24 h in Luria-Bertani (LB) medium. The *Chromobacterium violaceum* CV026, with the purple pigment violacein as a reporter [20], was purchased from ATCC and was most sensitive to the signaling molecules of C6-HSL and C4-HSL. The strain was cultivated at 28 °C and 150 rpm for 24 h in Luria-Bertani (LB) medium containing 30 mg/mL kanamycin. *A. hydrophila* ATCC 7966 was purchased from ATCC and was grown in Luria-Bertani (LB) medium at 30 °C for 20 h.

### 2.2. Plasmid and Reagents

*E. coli* DH5α, and the pGEM-T Easy vector were purchased from Promoga (Madison, WI, USA). *E. coli* Trans-1 and pEASY-Blunt E1 were bought from Transgen (Beijing, China). C6-HSL and C4-HSL were available from Sigma-Aldrich (St. Louis, MO, USA). Isopropyl-d-1-thiogalactopyranoside (IPTG) (50 mg/mL) and 5-bromo-4-chloro-3-indolyl-Dgalactopyranoside (X-gal) (20 mg/mL) were purchased from Promega (Madison, WI, USA). *PfuTaq*DNA polymerase and restriction endonucleases were bought from TaKaRa (Otsu, Japan). The DNA purification kit, T4 DNA ligase was bought from Invitrogen (Carlsbad, CA, USA). Other chemical reagents came in chemically pure grades and were commercially available fromTiangen (Tiangen & GreenFortune, Beijing, China).

### 2.3. Cloning and Sequencing of the AHL Lactonase Gene Ytnp

A bacterial genome extraction kit was used to extract genomic DNA of *B**. licheniformis* T-1 according to the instructions, which was used as the PCR amplification template. According to the conserved sequence of T-1 and known information, design specific primers: *ytnP*-F1: ATGAAGCTGATTCAGGTTGCATT; YtnP-R1: CATGCGGCTTTCTCTTTTTACTGAC. The PCR procedure consisted of 30 cycles at 98 °C for 5 min, 94 °C for 30 s, 55 °C for 30 s, 72 °C for 1 min, and finally 72 °C for 5 min. PCR products were ligated to the pGEM-T Easy carrier sequence. The nucleotide sequences and open reading frames were analyzed using Vector NTI 10 software and NCBI Open Reading Frame Finder (http://www.ncbi.nlm.nih.gov/gorf/gorf.htmL, accessed on 20 February 2018). The use of SignalP searched signal peptide (http://www.cbs.dtu.dk/services/TMHMM-2.0/, accessed on 20 February 2018). The DNA and protein sequences were compared with known sequences using the BLASTP and BLASTN procedures (http://www.ncbi.nlm.nih.gov/BLAST/, accessed on 20 February 2018). AHL esterase (Protein Data Bank accession number c3ehB) (98% sequence identity) was used as a template, using the SWISS-MODEL (http://swissmodel.expasy.org/, accessed on 21 February 2018) and SWISS-PdbViewerDeepView v 4.0 software to predict YtnP tertiary structure.

### 2.4. Expression and Purification of YtnP

The *B.*
*Licheniformis* t-1 *ytnP* gene PCR was used to expand the use of modified from genomic DNA primers and EcoRI restriction sites (5′3′, BD-*ytnP*-F1: ATGTTCGGCGTTGTTCCCA BD-*ytnP*-R1: CTAGCGGCTTTCTCTTTTTACTGAC); 40 cycles (5 min at 98 °C, 94 °C for 30 s, 55 °C for 30 s, 72 °C for 1 min, for the final 10 min incubation at 72 °C). The PCR DNA was directly purified with the Tianquick MIDI purification kit, which was digested with NDEI and EcoRI (NEB) enzymes, and ligated into the expression plasmid pEASy-Blunt E1(transgenic) with T4 DNA ligase to transform *Escherichia coli* BL21 (DE3) cells. Using the TIANprep miniature plasmid kit (Tiangen, Beijing, China), the plasmid DNA were isolated from the recombinant clone, and sequencing confirmed the insert pEASY-Blunt no.e1-YtnP. Next, the recombinant plasmid pEASY Blunt E1-YtnP into *E. coli* BL21 (DE3) cells, 37 °C in containing 50 mg/mL of ampicillin LB culture medium for the night. Positive transmitters were cultured in LB medium containing 50 mg/mL ampicillins at 37 °C for 4 h with an optical density of 0.6 at 600 nm (OD600). 1 mM IPTG induction in mixture at 37 °C was added, with shake cultivation in 150 rpm for 8 h.

The cells were harvested by centrifugation at 12,500× *g* for 15 min. A mixture of DNase I, lysozyme, and protease inhibitor was added according to the bacterial protein extraction kit. Purification YtnP and cell lysis liquid (5 mL) in Ni-NTA resin (1 mL; QIAGEN, Hilden, Germany) was balanced with a binding buffer (20 mM Tris-HCl of pH7.9, 5 mM imidazole, 0.5 mM NaCl, 8 mM urea). The proteins were eluted in the elution buffer with a linear gradient of 50–500 mM imidazole. The purified proteins were analyzed with SDS-PAGE (12%), and the apparent molecular weight was measured by using Coomassie Bright Blue R-250 molecular weight standard. The YtnP protein concentration was determined by using the Bradford method [10].

### 2.5. Bioassay of AHL-Lactonase Activity

The C6-HSL was used as a substrate to measure the activity of AHL-lactonase of purified YtnP protein. The 2 mL of overnight culture *C. Violaceum* CV026 was mixed with 200 mL LB agar and poured into a Petri dish. A sterile punch was used to make 5 mm diameter holes in the plate. AHL-lactase 200 μL reaction system consisted of 10 μL YtnP protein samples and 190 μL reaction solutions containing 24 nM of the final concentration of C6-HSL, which was placed in phosphate buffer with 50 mM (pH 8.0) and incubated at 25 °C for 45 min. The reaction was terminated by adding 50 μL 10% (*w*/*v*) SDS. The reaction mixture was then transferred into plate holes, and the radius of the *C. violaceum* CV026 non-pigmented zoned in 48 h was used to determine residual C6-HSL levels. AHL lactase activity reported on this paper is the degradation rate of AHL or the AHL degradation activity unit (U) per mg of protein, where 1U is the amount of enzyme required to hydrolyze 1 μM of C6-HSL per hour under the above conditions (mg L^−1^ h^−1^).

### 2.6. YtnP Inhibition of A. hydrophila Biofilm Formation

A 24-well round-bottom plate with a lid was used to investigate the impact on purified YtnP on *A. hydrophila* biofilm formation in vitro. *A. hydrophila* was incubated statically in a 24-well plate containing 2 mL of LB medium with various concentrations of purified YtnP (0–80 ng/μL) at 30 °C for 24 h. At the end of the incubation period, the planktonic cells (measured growth at 600 nm) and the medium were poured out. According to the scheme described by Junker [20], for better display, crystal violet staining was used in the experiment. In order to quantify the formation of biofilms, the crystal violet associated with biofilms was dissolved in 95% ethanol, and the spectrophotometry was used to determine the optical density of 595 nm [7]. Appropriate controls were maintained.
Inhibition (100%) = (*OD_control_* − *OD_YtnP_*)/*OD_YtnP_* × 100%

Note: *OD_control_* was the *OD*_590_ of *A. hydrophila*, *OD_YtnP_* was the *OD*_590_ at the presence of YtnP.

### 2.7. Determination of Minimum Inhibitory Concentration

Minimum inhibitory concentration (MIC) of four kinds of antibiotics such as enrofloxacin, florfenicol, thiamphenicol, and doxycycline hydrochloride was determined against *A. hydrophila* by microdilution method [8]. The consequence of the synergistic effect of purified YtnP and antibiotics on MIC of *A. hydrophila* was determined according to the method of and concentration of the purified YtnP was 80 ng/μL.

### 2.8. Virulence of A. hydrophila Inhibited with YtnP

The *aerA**, ahyB**, hem**, ast**, ep* of virulence gene of *A. hydrophila* were used to investigate the inhibition effect of YtnP. *A. hydrophila* was inoculated in 90 mLof LB broth at 10^5^ CFU/mL and co-cultured with purified YtnP without YtnP at 160 r/min at 30 °C. At the indicated time points after 4, 8, 12, 16, 20, 24, 30, 36, 48, and 60 h, the RNA of the *A. hydrophila* was sampled and extracted using a TRIzol RNA extraction kit (Invitrogen, Carlsbad, CA, USA) according to the manufacturer’s protocol. To ensure sufficient integrity of the extracted RNA for subsequent experiments, the optical density (*OD*) ratio *OD*_260_:*OD*_280_ was measured on a spectrophotometer (UV1600, Amershampharmacia biotech, INESA Analytical Instrument, Shanghai, China). The purity of RNA was detected by 1% gel electrophoresis. A primer mixture designed and synthesized using a reverse transcription kit (NovoScript^®^ 1st Strand cDNA Synthesis SuperMixc DNA, Shanghai, China) and Primer Premier 5.0 software was used for Figure 1B i DNA (cDNA); the specific primers of virulence gene of *A.*
*hydrophila* ATCC7966 is shown in Table 1. Using cDNA 2 μL, positive primers (1 μL, 8 mM), reverse primer (1 μL, 8 mM), NovoStrart^®^ SYBR qPCRSuperMix (10 μL, Jinan Co., Shanghai, China), and ddH2O, a mixture of 6 μL in the Bio-Rad iQ5 real-time PCR (Hercules, CA, USA) in the real-time polymerase chain reaction (rt-pcr). The RT-qPCR reaction was carried out under 38 cycles of 95 °C preheating for 15 s, 95 °C amplification for 15 s, and 60 °C amplification for 45 s. The final reaction volume was 20 μL. The amplification of b-submit RNA (a relatively unchanged reference gene) was carried out in parallel, and the number of cDNA was normalized to the equivalent *B-submit* mRNA level [9]. This experiment was carried out in three independent experiments. According to the number of known cDNA, generated a standard curve for each pair of primers. The relative mRNA expression was calculated by 2^ΔCT^ equations.

### 2.9. Effect of YtnP against Infection of A. hydrophila Infection

Health *Carassius auratus* of 100 ± 5 g were raised at the density of 10 fish/tank (volume: 5 L/tank) in an indoor recirculation aquaculture system with daily aeration. The water temperature was maintained at 28 ± 1 °C. *A. hydrophila* ATCC 7966 was cultured in LB media at 30 °C for 20 h and rinsed with sterile PBS buffer (pH 7.6) 3 times as the injection preparation. The purified YtnP was suspended in a PBS buffer with a pH of 7.6 for further use.

Duplicated groups of eight *C. auratus* (100 ± 5 g, 10 cm ± 2 cm) were injected into the abdominal cavity with 1000 μL of *A. hydrophila* ATCC 7966 cell suspension at 5 × 10^5^–5 × 10^10^ CFU/mL, and control groups were injected with the same volume of PBS (pH 7.3). Lethal lose 50% (LD_50_) of *A. hydrophila* ATCC 7966 on *C. auratus* for 96 h was determined by the injection-infection method. The experimental fish was anesthetized with MS-222 at 80 mg/L. The 96-h cumulative mortality was recorded, and all dead specimens were removed. LD_50_ was calculated by the Reed method and Muench method [21]. The challenge test was performed in three replica aquariums. To determine the effect of YtnP on *A. hydrophila* ATCC 7966 infection, *C. auratus* were randomly divided into four test groups injected intraperitoneally with 0.81 mg/mL YtnP (AHLase), 0.80 mg/mL YtnP + *A. hydrophila* of 10^8^ CFU (AHLase + *A. hydrophila*), *A. hydrophila* of 10^8^ CFU (*A. hydrophila*), and PBS buffer (control treatment) per fish in the volume of 1000 μL when anesthetized by tricaine methanesulfonate (MS-222). Each group had two replicates, with each replicate containing 10 fish. The system was aerated, the dead specimens were taken out every day for routine bacteriological examination. The cumulative mortality rate for each treatment was recorded every 2 h of 96 h. This procedure was approved by the Ethics Committee of the Author’s institution.

### 2.10. Nucleotide Sequence Accession Number

The nucleotide sequence of *N*-acyl isoserine lactonase gene (*ytnP*) from *B**. licheniformis* T-1 has been deposited in GenBank (No. MK360859).

## 3. Statistical Analysis

All the experiments were carried out in triplicate. Analyzed by SPSS 16.0 software, the data to mean ± standard deviation (SD). Using the single factor analysis of variance (ANOVAs) and multiscale comparison between Duncan’s test to determine the significant difference and to determine the significant level of α = 0.05.

## 4. Results

### 4.1. Gene Clone and Sequence Analysis of ytnp

The AHL lactase gene YtnP were cloned from *B. licheniformis* T-1 genomic DNA based on the primers YtnP-F1 and YtnP-R1 (Figure 1A). YtnP encoded 256 residues and a stop codon with a calculated molecular weight of 29.0 kDa and a PI of 5.62. According to analysis of SignalP 3.0, it was inferred that there was no signal peptide in the amino acid sequence of YtnP. Phosphorylation site FGVVPKPLWPSK was identified, which was consistent with previous reports [7]. The YtnP gene of T-1 was 99% homologous to the homologous gene of *B. subtilis* sub sp BSP1 by BLAST search using the AHL lactonase metal-β-lactamase gene. Homologous model showed that YtnP had a typical structure of metallolactamase (MBL) family proteins in comparison with AHL lactonase enzyme from *Bacillus* (PDB login c3ehB). From Figure 1B, it can be observed that YtnP is surrounded by a hydrophilic alpha helix, and the entire protein center was constituted by β-sheets inside.

The result of the Phylogenetic tree of YtnP indicated that YtnP protein belonged to the AidC cluster of the MBL family, the similarity of YtnP protein in T-1 was 100% in compassion with *B. subtilis* 3610 (Figure 2).

### 4.2. Expression and Purification of YtnP

In the expression study, the activity of AHL lactonase in cell lysate was detected at the end of the culture. Otherwise, no activity was detected in the supernatant of untransformed or transformed cells containing empty plasmids. The molecular weight of YtnP purified by SDS-PADG analysis was 36 kDa and higher than the predicted value (including part of His tag) 29 kDa (Figure 3). The optimal elution concentration of imidazole was 150 mM for the target protein. The AHL lactonase activity of the purified recombinant YtnP was 1.097 ± 0.7 U/mL with C6-HSL as the substrate.

### 4.3. YtnP Inhibition of A. hydrophila Biofilm Formation

Different concentrations of YtnP were added into the culture medium of *A. hydrophila* and the absorbance at *OD*_600_ nm was measured, YtnP had no effects on the growth of *A. hydrophila* in comparison with the control group (Figure 4A). There was no significant inhibition of growth (*p* > 0.05). The results revealed that the development of biofilm was much more affected by quorum sensing interference than the growth performance. A significant level of biofilm-inhibition was observed and measured by crystal violet staining at a gradient dilution in all the concentrations, including 80, 40, 20, 10, 5, 2.5, 1.25, 0.625, 0.315, and 0.15625 ng/μL compared with control. At a concentration of 80 ng/μL, YtnP showed a maximum of 68% reduction in the biofilm forming capability of *A. hydrophila*. The 2-fold dilution of YtnP concentration on 40 ng/μL or 20 ng/μL, inhibition of *A. hydrophila* biofilm beyond 50%, and when the concentration of YtnP dropped below 5 ng/μL, the inhibition also reached more than 30%. Thus, YtnP had a significant inhibitory effect on the formation of BF in *A. hydrophila* (*p* < 0.05).

### 4.4. Synergistic Effect of YtnP and Antibiotics on MIC of A. hydrophila

MIC of the combination of antibiotics and purified YtnP on *A. hydrophila* were measured by *OD*_600_ r MIC of doxycycline hydrochloride against *A. hydrophila* was 0.5 μg/mL and 0.125 μg/mL with or without YtnP, and MIC of thiamphenicol against *A. hydrophila* was 4 μg/mL and 1 μg/mL with or without YtnP (Figure 5). The growth curve showed there was no significant change in *OD*_600_ between treated and untreated in the groups of enrofloxacin and florfenicol cultures.

### 4.5. Virulence Gene of A. hydrophila Inhibited with YtnP

In the present study, the mRNA expression of *aerA, ast, hem, ep*, and *ahyB,* five virulence genes in *A. hydrophila* were measured in order to evaluate the effects of YtnP on those virulence mediated by QS. The results indicated that the gene expression quantity in *A. hydrophila* of the YtnP-treated group was significantly lower than that control groups (*p* < 0.05). For aerA, ast, and hem, there was a significant decrease in the mRNA expression of 60 h compared with the corresponding control groups (*p* < 0.01) (Figure 6). For ep, there was a significant decrease in the mRNA expression within 60 h except for 8 h compared with the corresponding control groups (*p* < 0.01). For *ahyB*, there was a significant decrease in the mRNA expression of 48 h compared with the corresponding control groups (*p* < 0.01). The results indicated that the *N*-acyl homoserine lactonase YtnP could down-regulate the expression of virulence genes in *A. hydrophila*.

### 4.6. Co-Injection of Recombinant YtnP and A. hydrophila in C. auratus

The virulence of *A. hydrophila* ATCC 7966 in *C. auratus* reared under standard aquaculture conditions was confirmed by abdominal injections, with individuals showing signs of tissue swelling, necrosis, ulceration, and hemorrhagic septicemia and eventual death. The LD_50_ of *A. hydrophila* ATCC 7966 was 5 × 10^8^ CFU/mL. In protection experiments, no pathogenic symptoms and mortalities were observed when intraperitoneal injection of sterile PBS buffer (groups A) or YtnP alone (Group B) for 96 h, the results suggested that the PBS and YtnP were not toxic to the fish. When injected intraperitoneally with *A. hydrophila* + YtnP, the accumulated mortality at 96 h was 27 ± 4.14%, significantly lower than the average mortality of 78 ± 2.57% of the *C. auratus* injected with 10^8^ CFU/mL of *A. hydrophila* (*p* < 0.001) (Figure 7). It indicated that YtnP significantly attenuated *A. hydrophila* infection on *C. auratus*. Co-injection of YtnP and *A. hydrophila* decreased the mortality rate of *C. auratus* by nearly 50%. The mortality of fish injected with *A. hydrophila* only was 30% at 16 h, and there was no fish dead within 16 h at the group of *A. hydrophila* + YtnP.

## 5. Discussion

Anti-virulence therapy methods have gained more and more attention against pathogenic bacteria infection as a new antimicrobial strategy in the situation of increasing from drug resistance [2]. AHL degrading enzymes have been documented before as a promising way to prevent and control bacterial diseases in the agriculture and aquaculture industries [22,23]. As one of the quorum quenching genes, *ytnP* was discovered firstly in *B.subtilis*, and until recently, it was reported that the *ytnP* gene had been found in *B. licheniformis* T-1 in our study [10,24]. In the present experiment, the *ytnP* gene from *B. licheniformis* T-1 was cloned and expressed in the *E. coli*. It was the first report on the *N*-acyl homoserine lactonase YtnP from *B**. licheniformis* in this experiment. The result of the phylogenetic tree based on the amino acid of YtnP implied that YtnP belonged to the AidC cluster of the MBL family, which was very close to the YtnP from *B**. subtillis* 3610 and have a certain relationship in comparison with AiiA cluster (Figure 2). Otherwise, VinojG et al. cloned the aiiA gene in *B. licheniformis* DAHB1 [23], which was not consistent with the results that we have found in this study. We have even tried to clone the aiiA gene in the *B. licheniformis* T-1, but failed. Although the aiiA gene was found in multiple *Bacillus* spp. [25,26], there were not direct evidences to prove that more than two kinds of quorum quenching genes existed in one microorganism species. As far as the quorum sensing system, it was ever reported that the QS in *P. aeruginosa* was well studied with 3 major hierarchial complex signaling systems, such as LasI-LasR, RhlI-RhlR and PQS-MvfR systems, accounting for the production of several virulence [27]. To be the truth, we still did not have idea whether the quorum quenching systems in *Bacillus* spp. were like the quorum sensing systems in *P. aeruginosa*, which there were more than one signaling interfering systems. Based on the discovery of AiiA and YtnP in *B. licheniformis*, we hypothesized that there might be more than two kinds of quorum quenching systems in *B. licheniformis* involved in the interactive relationship of the microorganism in the natural environment, which need further researches.

The Gram-negative pathogens of *Aeromonas* spp. use AHLs as signaling molecules [28]. Biofilm formation, and expression of virulence factors are mediated by AHLs has been reported in *Aeromonas* spp. before [29,30,31,32]. And so, a quorum-quenching strategy might efficiently control these pathogens against infection. In the present study, we monitored the antimicrobial effect of AHL homology YtnP protein. Firstly, the inhibition effect of YtnP on *A. hydrophila* biofilms formation was quantified by crystallographic staining method. And the result was consistent with previous documents that YtnP expressed in *B. subtilis* added to cultures of *P. aeruginosa* could inhibit biofilm formation [9]. Meanwhile, Brackman et al. synergistically acted on substances with QS inhibition ability and found that QS inhibitors could increase the sensitivity of bacteria to antibiotics [15]. In this study, the synergy of purified recombinant YtnP protein and antibiotics significantly reduced MIC of doxycycline hydrochloride, thiamphenicol (Figure 5A,C) (*p* < 0.05). The QS signal molecule has been shown to mediate biofilm formation of Gram-negative bacteria to protect the host against antibiotics, the drug resistance produced by bacterial biofilms was a systematic and complex drug resistance mechanism, which reduced application effects of antibiotics in the control of aquaculture diseases [32]. As a quorum-sensing quenching enzyme, the recombination YtnP protein could interfere with the bacterial communication by degrading the quorum-sensing signal molecular C6-HSL and break the development of *A. hydrophila* biofilm. Moreover, antibiotics could be used in combination with YtnP, which did not adversely affect the use of YtnP activity and avoided the drawbacks that quorum quenching probiotics could not be used simultaneously with antibiotics. While there were no significant effects on reducing the MIC on enrofloxacin and florfenicol with the synergy of YtnP and antibiotics. We speculated that the interference with biofilm formation could decrease the doses of the antibiotics, there are still differences contraposed to different antibiotics and which need more investigation to explore the working mechanism. Secondly, the QS system regulated the production of virulence factors of *A. hydrophila* [33]. With the help of the virulence factors, the pathogenic bacteria could break the host immune defense and invade the organism body [24]. Thus, to interfere with the QS system, reduce the expression of virulence factors could control the occurrence of aquatic diseases caused by *A. hydrophila* [27,34,35]. As we know, five virulence genes *aerA, hem, ahyB, ast* and *ep* were essential to the pathogenicity of *A. hydrophila*. In the present study, YtnP could decrease mRNA expression of five virulence genes in *A.hydrophila* significantly and reduce the pathogenicity of *A. hydrophila* (*p* < 0.01). The mRNA expression of the five genes above was decreased significantly in 60 h in comparison with the controls. The results indicated that the *N*-acyl homoserine lactonase YtnP could down-regulate the expression of virulence genes, which was significantly higher than that report of Chen [36]. The in vitro results were also a response to the of challenge test in vivo, when the *C. auratus* were intraperitoneally injected with *A. hydrophila* ATCC 7966 and YtnP, the accumulated mortality at 96 h was 27 ± 4.14%, which was significantly lower than the average mortality of 78 ± 2.57% of the *C. auratus* injected with 10^8^ CFU of *A. hydrophila* in control groups (*p* < 0.001) (Figure 7).

Hence, YtnP enzyme, both in vivo and in vitro, had shown that it could enhance the protective effect on aquatic animals infected with *A. hydrophila* by attenuating the expression of virulence factors. In addition, after injected with the YtnP enzyme alone, the *C. auratus* displayed no signs of stress or disease, and no mortalities were observed (Figure 7), which indicated that the YtnP enzyme was safe from applications.

Especially, survival of the group injected with the combination of YtnP and *A. hydrophila* ATCC 7966 was 100% at 0–16 h in the present experiment; otherwise, the death of the control group injected with *A. hydrophila* ATCC 7966 only occurred at the 4th hour, which was significantly earlier than the experimental group. Based on this analysis, we can speculate that the best protection stage of the YtnPprotein against infection of *A. hydrophila* might be in the early infection, which suggested that YtnP protein could be used as the prophylactic agents in aquaculture practices.

Anyway, these results indicated that YtnP, as the quorum quenching enzyme, could reduce the AHL accumulation of *A. hydrophila*, which would affect related gene expression and reduce the pathogenicity against *A. hydrophila* infection. YtnP as a *N*-acyl homoserine lactone enzyme, had great potential application value on prevention and control bacterial diseases by way of anti-virulence.

## Figures and Tables

**Figure 1 antibiotics-10-00631-f001:**
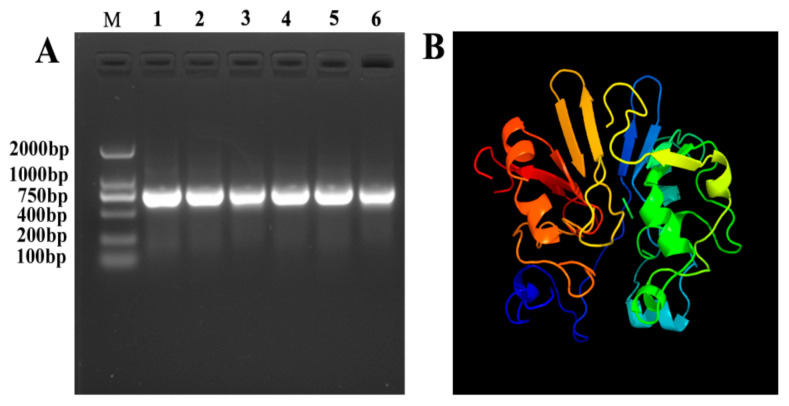
(**A**) PCR amplification of *ytnP* gene. Lane M: 2000 bp DNA Ladder (Invitrogen), Lane 1–6: *ytnP* gene (~770 bp). (**B**) The putative tertiary structure of YtnP modeled using the crystal structure of AHL lactonase from *Bacillus* (PDB accession number c3ehB) as the template. The coiled structure represents the alpha helix and the arrow represents the beta sheet.

**Figure 2 antibiotics-10-00631-f002:**
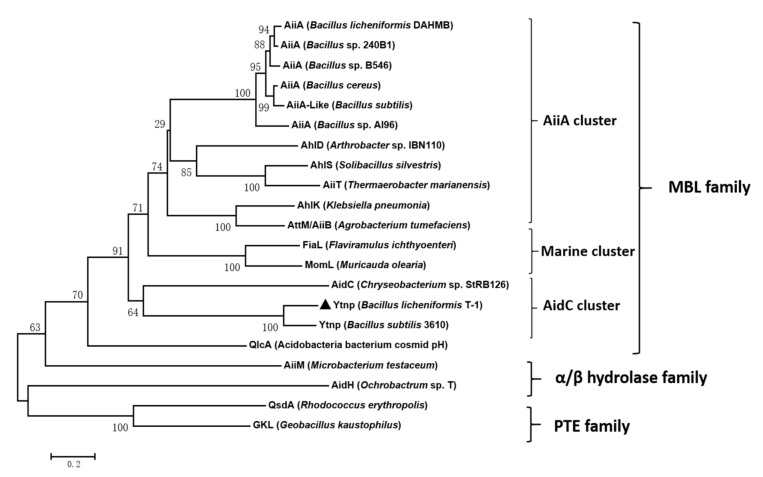
The phylogenetic tree was constructed according to the amino acid sequences of YtnP. The AHL lactonases and AhlD from *Arthrobacter* sp. (AAP57766.1), AhlK from *K. pneumoniae* (AAO47340.1), AhlS from *Solibacillus silvestris* (BAK54003.1), AidC from *Chryseobacterium* sp. strain StRB126 (BAM28988.1), AidH from *Ochrobactrum* sp. T63 (ACZ73823.1) *Bacillus* sp. strain 240B1 (AAF62398.1), *Bacillus* sp. (AAF62398.1), *Bacillus* sp. strain AI96 (HM750248.1), AiiM from *Microbacterium testaceum* strain StlB037 (YP_004225655.1), AttM/AiiB from *A. tumefaciens* (AAL13075.1), QlcA from Acidobacteria bacterium cosmid p2H8 (ABV58973.1), QsdA from *Rhodococcus erythropolis* (AAT06802.1), AiiT from *Thermaerobacter marianensis* (AB935246.1), FiaL from *Flaviramulus ichthyoenteri* Th78T (WP_034041734.1), MomL from *Muricauda olearia* (AIY30473), and GKL from *Geobacillus kaustophilus* (WP_011231002.1). These proteins were classified as the member of the metallo-β-lactamase family (MBL), α/β-hydrolase family, and phosphotriesterase family (PTE). The dendrogram was constructed by the neighbor-joining method using Clustal W. The scale bar represents 0.1 substitutions per amino acid position. These proteins were classified as the member of the metallo-β-lactamase family (MBL), α/β-hydrolase family, and phosphotriesterase family (PTE). The dendrogram was constructed by the neighbor-joining method using Clustal W. The scale bar represents 0.1 substitutions per amino acid position.

**Figure 3 antibiotics-10-00631-f003:**
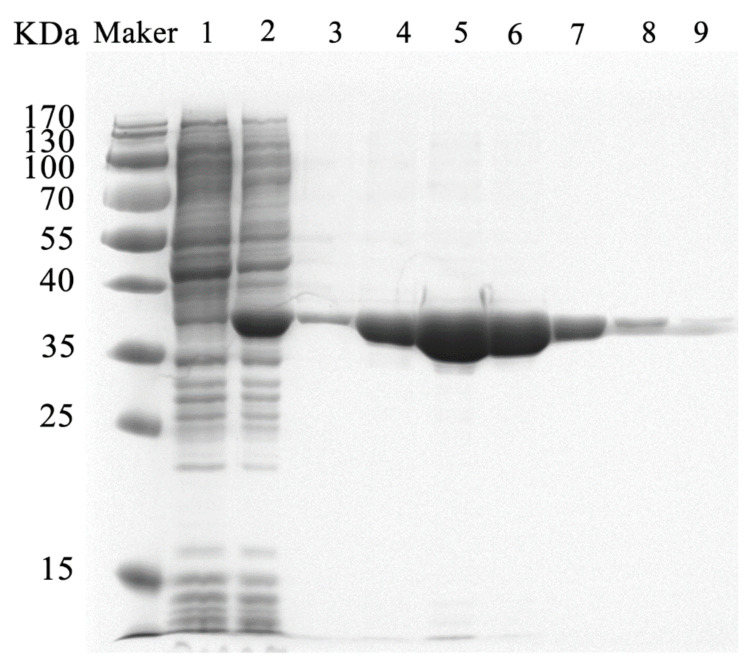
SDS-PAGE gel of recombinant YtnP from *E.coli* BL21 (DE3). Lanes: M, protein molecular mass markers; 1, cell extract of *E. coli* BL21 (DE3) that harbored pEASY-Blunt E1 carrying *ytnP* and that was not treated with IPTG; 2, cell extract of *E. coli* BL21 (DE3) that harbored pEASY-Blunt E1 carrying *ytnP* and that was treated with IPTG; 3–9, purified YtnP with different concentration of imidazole elution, 50, 100, 150, 200, 300, 400, 500 nM.

**Figure 4 antibiotics-10-00631-f004:**
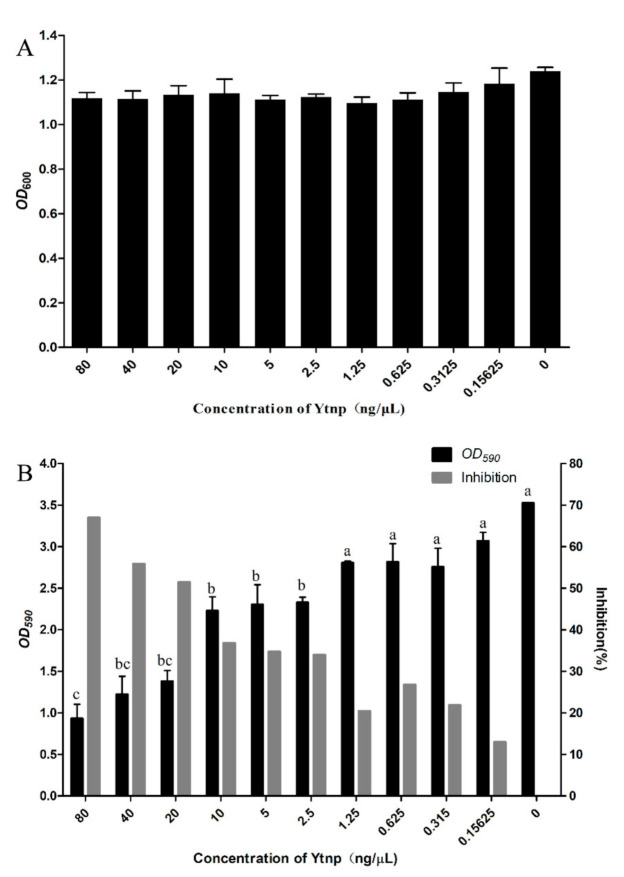
(**A**) Effect of YtnP on the growth of *A. hydrophila*. (**B**) YtnP inhibited biofilm formation of *A. hydrophila* based quorum sensing interference on a polystyrene surface. The same letter (a–c) represent no significant difference (*p* < 0.05).

**Figure 5 antibiotics-10-00631-f005:**
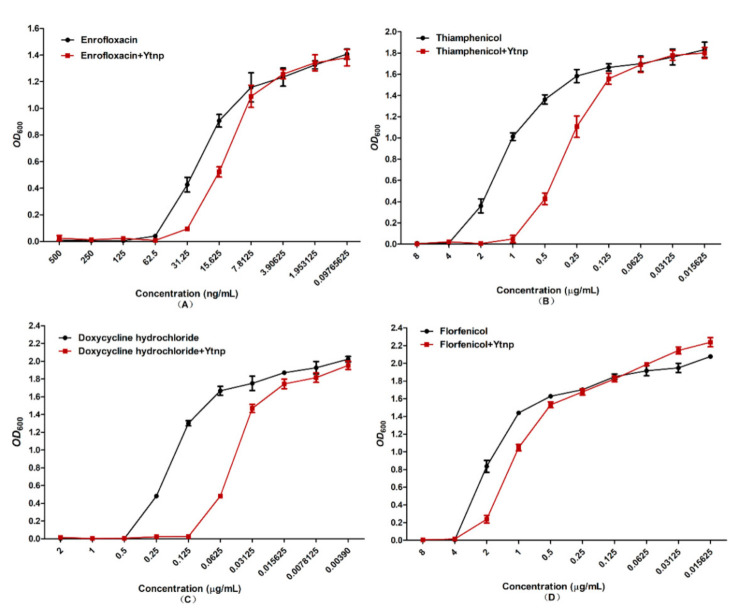
Synergistic effect of YtnP and antibiotics on MIC of *A. hydrophila*. (**A**) Enrofloxacin, (**B**) thiamphenicol, (**C**) doxycycline hydrochloride, (**D**) florfenicol.

**Figure 6 antibiotics-10-00631-f006:**
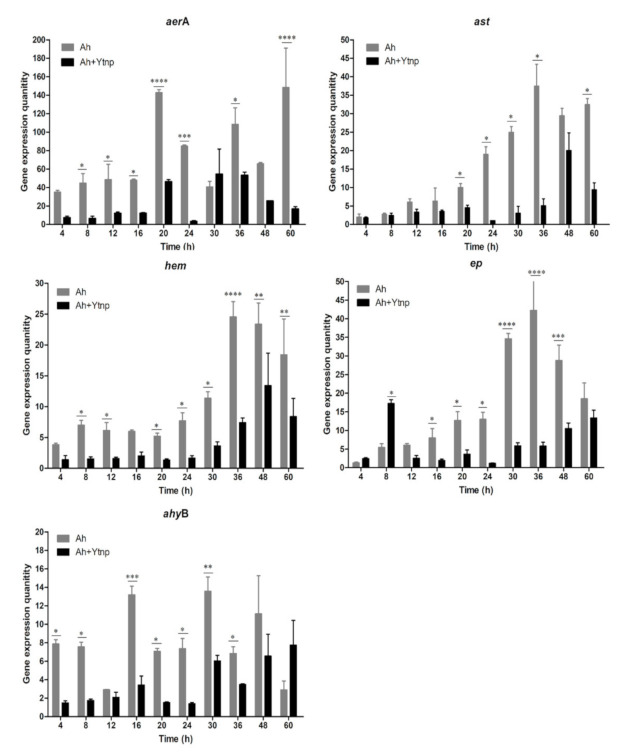
Quantification of virulence genes *aerA*, *ast*, *hem*, *ep*, and *ahyB* (Five virulence genes aerA, hem, ahyB, ast and ep were essential to the pathogenicity of *A. hydrophila*) in *A. hydrophila* ATCC 7966 at indicated time points (4, 8, 12, 16, 20, 24, 30, 36, 48, and 60 h) after adding AHL lactones YtnP. The “*” presents that there was a decrease in the mRNA expression in A. hydrophila ATCC 7966 of the YtnP-treated groups compared with the corresponding control group (*p* < 0.05); and the “**, ***, ****” present that there was a significant de-crease in the mRNA expression in *A. hydrophila* ATCC 7966 of the YtnP-treated groups compared with the corresponding con-trol group (*p* < 0.01).

**Figure 7 antibiotics-10-00631-f007:**
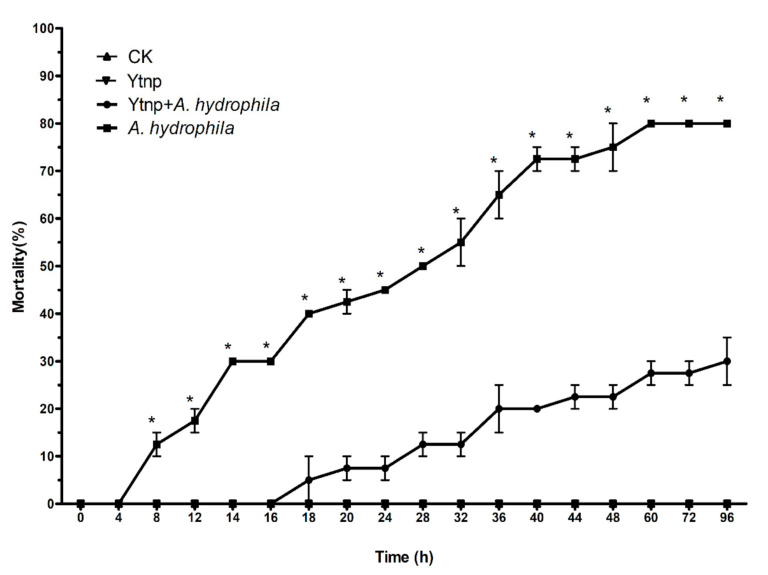
Mortality of *C. auratus* injected intraperitoneally with *A. hydrophila* ATCC 7966 or *A. hydrophila* ATCC 7966 and YtnP. Groups were as follows: Group A (CK) was injected with sterile PBS buffer, Group B (YtnP) was injected with AHLase YtnP, Group C (YtnP + *A. hydrophila*) was injected with *A. hydrophila* ATCC 7966 + AHLase YtnP, Group D (*A. hydrophila*) was injected with *A. hydrophila* ATCC7966. Each value is mean ± SD (*n* = 2), data marked with “*” means significant difference (one-way ANOVA; *p* < 0.001) among groups.

**Table 1 antibiotics-10-00631-t001:** The specific primers of virulence gene of *A. hydrophila* ATCC7966 and reference gene.

Gene	Primer Sequence	Fragment Size
*B-subumit*	B-subumitF:gtgcgtgaaggtctgattgccgt B-subumitR:gcttctcacccatcgcctgttcg	160
*ast*	Ast-F:ccccgcctggctgttcttt Ast-R:cggcgaagtcttgcggtga	166
*aerA*	AerA-F:cctgagcctgtctgaccaagt AerA-R:gctcgggtcgaagttctcg	192
*hem*	hem-F:gtcatgacctgacgctgag hem-R:ctggtaacgaatgctgctc	137
*ep*	Ep-F:tcccattgccctgttgct Ep-R:tcgtcactgttgccatcca	122
*ahyB*	AhyB-F:gccgctgaatccctcctcctac AhyB-R:tgctgccgacgttgttcttgtag	156

## Data Availability

The data presented in this study are available on reasonable request from the corresponding author.

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
