# Peer review of "Attenuation of Aeromonas hydrophila Infection in Carassius auratus by YtnP, a N-acyl Homoserine Lactonase from Bacillus licheniformis T-1"

_antibiotics, 2021, doi:10.3390/antibiotics10060631_

Round 1

Reviewer 1 Report

The language and grammar used throughout are a significant limitation to readability. Figure 4B illustrates the findings well but could use a bit more clarity on the labelling.

Reviewer 2 Report

The study on the Attenuation of Aeromonas hydrophila infection in Carassius auratus by YtnP, a N-acyl homoserine lactonase from Bacillus licheniformis T-1 has been carried very methodically by  the authors. They first characterised the YtnP enzyme and cloned it  to produce enough for further studies. These studies clearly showed its role as a quorum quencher which can increase the potency of some antibiotics. More importantly  they showed its potential in reducing biofilm production  by the pathogen which results in a slower infection. Overall I am pleased with the outcomes of the study. However there are too many typographical and grammatical errors which calls for a major editing job to be undertaken.

Reviewer 3 Report

The manuscript can be accepted after minor revisions in abstract and introduction. Abstract needs to be short and concise. Introduction should include the prior studies on A. hydrophila.

Round 2

Reviewer 1 Report

The results and materials and methods are improved with results to grammar and formatting, but significant editing of the introduction would add interest and more effectively engage the reader.

Author Response

pleas see the attachement